CropGCNN: color space-based crop disease classification using group convolutional neural network

Ahmad Naeem 1 nahmad.mca@nitrr.ac.in
Singh Shubham 1
AlAjmi Mohamed Fahad 2
Hussain Afzal 2
http://orcid.org/0000-0002-3646-6828 Raza Khalid 3 kraza@jmi.ac.in
1 Department of Computer Applications, National Institute of Technology Raipur (NITR) , Raipur , India
2 Department of Pharmacognosy, College of Pharmacy, King Saud University , Riyadh , Saudi Arabia
3 Department of Computer Science, Jamia Millia Islamia , New Delhi , India
Balas Valentina Emilia
Electronic publication date: 2024 Jul 29
Publication date: 2024
Volume: 10
Electronic Location ID: e2136
Received 2024 Jan 24; Accepted 2024 May 28
Copyright: © 2024 Ahmad et al.
Copyright year: 2024
Copyright holder: Ahmad et al.
License: This is an open access article distributed under the terms of the Creative Commons Attribution License, which permits unrestricted use, distribution, reproduction and adaptation in any medium and for any purpose provided that it is properly attributed. For attribution, the original author(s), title, publication source (PeerJ Computer Science) and either DOI or URL of the article must be cited.
License URL: https://creativecommons.org/licenses/by/4.0/

Keywords: Convolutional neural network, Crop disease classification, Image processing, Image classification, Color space

Funding: King Saud University, Riyadh, Saudi Arabia RSP2024R122 Mohamed Fahad Alajmi and Afzal Hussain received financial support from the project number (RSP2024R122), King Saud University, Riyadh, Saudi Arabia. The funders had no role in study design, data collection and analysis, decision to publish, or preparation of the manuscript.

==============================
Classifying images is one of the most important tasks in computer vision. Recently, the best performance for image classification tasks has been shown by networks that are both deep and well-connected. These days, most datasets are made up of a fixed number of color images. The input images are taken in red green blue (RGB) format and classified without any changes being made to the original. It is observed that color spaces (basically changing original RGB images) have a major impact on classification accuracy, and we delve into the significance of color spaces. Moreover, datasets with a highly variable number of classes, such as the PlantVillage dataset utilizing a model that incorporates numerous color spaces inside the same model, achieve great levels of accuracy, and different classes of images are better represented in different color spaces. Furthermore, we demonstrate that this type of model, in which the input is preprocessed into many color spaces simultaneously, requires significantly fewer parameters to achieve high accuracy for classification. The proposed model basically takes an RGB image as input, turns it into seven separate color spaces at once, and then feeds each of those color spaces into its own Convolutional Neural Network (CNN) model. To lessen the load on the computer and the number of hyperparameters needed, we employ group convolutional layers in the proposed CNN model. We achieve substantial gains over the present state-of-the-art methods for the classification of crop disease.

Introduction

Image classification, a fundamental pillar of computer vision, delves deep into the realm of color imagery, primarily characterized by the ubiquitous RGB format. To the human eye, these digital images manifest as visually rich compositions; however, to a computer, they are a mere confluence of numeric data, devoid of intrinsic semantic meaning (Kekre et al., 2013). In contemporary image classification, conventional wisdom dictates the direct processing of images in their native RGB garb. Yet, this research embarks on an ambitious voyage, venturing into the uncharted territories of diverse color spaces, unfurling a novel paradigm poised to redefine the art of image classification (Nasrin et al., 2020; Rimiru, Gateri & Kimwele, 2022). At the heart of this exploration lies the concept of the color space—an intricately structured framework that orchestrates the translation of colors into both digital and analog representations. It can be envisioned as an abstract mathematical model, one that meticulously quantifies colors through numerical values (van den Berg et al., 2020). The RGB color space, a stalwart in the color model pantheon, is fundamentally system-dependent and conventionally encoded in 24 bits, with each triumvirate channel—Red, Green, and Blue—receiving an 8-bit allocation. This architectural choice bequeaths a channel value spectrum extending from the stark void of 0 to the radiant zenith of 255, underpinning the assumption that the entire spectrum of colors can be birthed through the chemical blending of red, green, and blue (Ajmal et al., 2018).

Going beyond the customary precincts of RGB, this expedition embarks on an illuminating odyssey through a constellation of alternative color spaces, among them, the enigmatic hue saturation value (HSV), the versatile LAB, the captivating YUV, the nuanced YCrCb (Luma (Y), Chrominance Red (Cr), Chrominance Blue (Cb)), the alluring LUV, the enigmatic XYZ, and the harmonious HLS. Take, for instance, the HSV color space—a construct finely attuned to the quirks of human color perception, delineating colors in a triptych of attributes: hue, saturation, and value. Hue meanders along the full circle of 360 degrees, saturations vacillate between the extremes of 0 to 255, while brightness or value dances between 0 and 255 (Ajmal et al., 2018). This research embarks on the journey to unearth the latent treasures concealed within these diverse color spaces, an exploration that holds the promise of revolutionizing the domain of image classification. It is important to recognize that within the expansive realm of color spaces exists a vast uncharted territory left unexplored. At its core, this research hinges on the profound realization that, from the computer’s unfeeling perspective, an image is nothing but a symphony of numerical values. Therefore, any transformation imposed upon these images emerges as a novel vista in the computer’s perception. The transformation of an image into distinct color spaces begets a cornucopia of fresh visual interpretations as perceived by the computer (Reinhard & Pouli, 2011). This innovative approach harnesses the potency of compact neural networks to classify images within these distinct color spaces, culminating in the fusion of the final layer from each network. This harmonious amalgamation delivers an accuracy that comprehensively encapsulates the multifaceted perspectives of the different color spaces involved (Street, 2010). This unification requires that the individual outputs remain as orthogonal as possible—a critical aspect that we shall meticulously elucidate in the forthcoming sections within the proposed approach.

This study aims to offer comprehensive guidelines for the successful implementation of classification tasks in the field of crop disease detection and classification. In computer vision and crop disease prediction, classification is a widely employed methodology for addressing various challenges. In recent years, numerous deep learning (DL) models have been introduced and utilized for classification, segmentation, and detection tasks within the domain of crop disease prediction, demonstrating notable improvements in performance compared to preexisting methodologies (Wongchai et al., 2022; Lakshmanarao, Babu & Kiran, 2021; Ferentinos, 2018). In crop disease prediction, high-resolution cameras are employed to acquire images of substantial dimensions, which exhibit notable disparities in size and characteristics (such as texture and color) compared to the images utilized for analysis in computer vision applications. Given the potential nature of computational crop disease as a research field, it is imperative to thoroughly investigate the influence of various color spaces on the classification of diseased images. This article examines seven distinct color spaces, namely RGB, HSV, LAB, YCrCb, XYZ, HSL, LUV, and YUV, in the context of crop disease classification tasks. The images of different crops in different color spaces are shown in Fig. 1. The study focuses on using different color spaces to assess the group convolutional neural network (GCNN) model. The PlantVillage (Hughes & Salathé, 2015) database is utilized for the purpose of evaluating the model in this particular implementation.

Figure 1 Images of different crops in different color spaces.

Related works

Image classification stands as a pivotal cornerstone within the vast domain of computer vision, and its pertinence has witnessed a notable upsurge in recent times, owing in large part to the emergence of influential datasets such as Imagenet (Deng et al., 2009), CIFAR (Recht et al., 2018), SVHN (Yang et al., 2021), CALTECH (Zhang, Benenson & Schiele, 2017), and a myriad of others. However, the true catalyst for transformative advancements in image classification has been the advent of deep convolutional neural networks, which have wrought a profound revolution in the realm of accuracy achievable by image classification algorithms. These breakthroughs, in turn, have laid the foundation for remarkable feats in diverse image classification tasks (Ahmad, Khan & Singh, 2023). Contemporary research endeavours resoundingly echo the sentiment that the key to attaining unparalleled accuracy lies in the augmentation of network depth. Indeed, current pinnacles in image classification, especially with formidable datasets like Imagenet, predominantly rely on architectures of extraordinary depth. Deeper networks have defied expectations and unveiled their remarkable prowess in navigating the labyrinth of intricate computer vision challenges (Lee & Kwon, 2017). While network depth undeniably is key to unlocking superior results, the path to enhancing network performance is far from linear (Rawat & Wang, 2017).

In response to the unique challenges posed by deeper networks, most notably the enigmatic vanishing gradient problem, novel architectural paradigms have been introduced to the discourse. The introduction of identity connections, prominently exemplified in pioneering networks like ResNets and Highway networks, has unfettered the flow of signals between layers, offering a lifeline for gradients at risk of vanishing (Gowda & Yuan, 2019). FractalNets, on the other hand, have charted a distinct course, amassing substantial depth by interweaving multiple parallel layer sequences while preserving a multitude of shorter paths within the network (Larsson, Maire & Shakhnarovich, 2016). The underlying principle that binds these diverse approaches is the establishment of short, direct paths connecting the early layers of a network to their distant counterparts, thus facilitating the seamless flow of information. Densenet, a departure from the prevailing norms, orchestrated a symphony where each layer disseminated its insights to every other layer, culminating in resounding success in image classification tasks (Gowda & Yuan, 2019). While these cutting-edge approaches have unfailingly delivered stellar results, they have done so at the cost of swelling the parameter count to unprecedented levels. This serves as a clarion call for further exploration, urging us to seek avenues where the quest for classification accuracy harmonizes with the imperative of resource efficiency, especially in resource-constrained environments.

Beyond the structural innovations, it is imperative to underscore that these approaches have conventionally harnessed images in their raw, unaltered state for classification tasks. In the context of our research, we venture beyond the conventional by propounding an innovative approach: the transformation of images through the prism of diverse color spaces. The preprocessing input via color conversion is not novel. It is a fertile ground for exploration. For instance, prior works have delved into utilising the YCrCb color space for skin detection (Phung, Bouzerdoum & Chai, 2002), color space YCgCr for face-detection (De Dios & Garcia, 2003), hybrid color spaces for pixel classification (Maktabdar Oghaz et al., 2015), and synthesis of color spaces for the nuanced analysis of soccer images (Stein et al., 2017). To scrutinize the impact of color space conversion, we have undertaken a comprehensive analysis focusing on skin detection, where the RGB color space emerged as the undisputed champion. In Khan et al. (2012), the study based on the Bayesian model indicates that LAB is an optimal choice for accurate skin pixel classification. With their divergent outcomes, these antecedent works underscore the intricate interplay between image content and the color space it represents. In all these recent studies, the images in their original color space are directly fed to the adopted models for the task of classification. In this article, images are transformed into different color spaces before applying the GCNN model. Our primary innovation revolves around amalgamation, where we combine multiple color spaces to orchestrate a symphony of enhanced accuracy, capitalizing on the inherent lack of strong correlation between these color spaces.

This mosaic of innovative approaches, the quest for depth, the triumph over vanishing gradients, the orchestration of novel architectures, and the art of combining color spaces, collectively propel the field of image classification into a promising future, one where accuracy and efficiency coexist in harmonious balance. This study introduces the utilization of diverse color spaces and utilizes a group convolutional layer for the purpose of extracting visual features in convolutional neural networks (CNN) to facilitate similarity matching. The GCNN utilizes three blocks of different layers to attain optimal performance. The present study provides a concise summary of its contributions in the following manner: 1. The study successfully determines the best color space for classification tack using the proposed GCNN model.

2. The study achieved the highest level of performance concerning different parameters of GCNN.

3. The study successfully demonstrates the efficacy of several color spaces using straightforward group convolutional in the proposed. It is observed that various color spaces exhibit varying degrees of effectiveness in retrieving different types of images in the PlantVillage (Hughes & Salathé, 2015) dataset. As a result, it is advisable to perform color transformations during the image extraction process to get optimal retrieval outcomes.

Proposed gcnn model

The concept of our study originated from an investigation into the effects of color space conversions on the PlantVillage dataset. This investigation is done on a proposed neural network model constructed with great expertise using the Keras Sequential API. It has been carefully designed to address the complex problem of crop-disease classification with different color spaces. The model’s journey unfolds with an input layer, poised to receive images with dimensions of 32 × 32 pixels and three color channels, as is customary in image analysis. The GCNN model has been developed with the objective of minimizing the processing resources needed. The model architecture is shown in Fig. 2. The model is created based on the design principles of ResNet (Ahmad, Gupta & Singh, 2022). The number of residual blocks (N, which is a multiple of 2) in each stage can be adjusted to construct a GCNN model with varying depths. Here, two residual blocks are taken between the processing block and classification block to obtain the simple GCNN model. These three blocks of the model are explained in the next subsections.

Figure 2 Architecture of the proposed GCNN.

Processing block

In this block, group convolutional, batch normalization, parametric ReLU (PReLU), and average pooling layers are used to process the input image of different color spaces. The very first layer is a group convolutional layer adorned with 54 filters, each with dimensions of 3 × 3. It employs a ‘groups’ parameter set to three, ushering in the era of group convolution, while ‘padding’ is carefully set to ‘same’ to usher in the world of zero-padding, meticulously preserving the spatial dimensions of the input images. To lessen the load on the computer, computation, and response time, the model must be lightweight. This is achieved at three chosen as the optimal “groups” parameter for the proposed model. The renowned ReLU activation function is then summoned into action, infusing the model with non-linearity, followed by a touch of Batch Normalization, a critical stabilizing influence on the training process. In the relentless pursuit of model expressiveness, the subsequent introduction of a Parametric Rectified Linear Unit (PReLU) activation stands as a defining moment. In stark contrast to the conventional ReLU, PReLU introduces learnable parameters, unshackling the model’s ability to encapsulate intricate patterns in data. A strategic AvgPooling layer, boasting a 2 × 2 window, enters the scene, effectively downsizing the spatial dimensions, a crucial feature in managing model complexity.

Residual block

The narrative of the model unfolds further with the inception of one residual block, a pivotal construct comprising one group of convolutional layers, each generously equipped with 108 filters, each with dimensions of 3 × 3. The fundamental concepts of group convolution and batch normalization, which are essential for ensuring stable training, remain steadfastly followed. After the convolution process, the PReLU activation functions serve as protective mechanisms, similar to sentinels, that regulate the flow of non-linear transformations, thereby enabling the extraction of intricate features.

Classification block

Evolving in the spirit of dimensionality reduction, a MaxPooling layer emerges, wielding a 2 × 2 window, effectively reducing the spatial footprint of the model. The climax of the model’s convolutional journey is punctuated by a 1 × 1 convolutional layer, housing 108 filters and adorning group convolution with the venerable ReLU activation. With ‘padding’ set to ‘same’, this layer acts as a bridge between the convolutional and fully connected domains of the model. The transition from a 2D feature map to a 1D vector is eloquently executed through the model’s flattening mechanism, preparing the model for the impending fully connected layers. Entering fully connected layers, the transition is initiated by a dense layer comprising 128 neurons, which is accompanied by the utilization of the ReLU activation function. In order to overcome the risks associated with overfitting, a dropout layer is incorporated into the model with a carefully chosen rate of 0.5, serving as a diligent guardian of the model’s ability to generalize. Following the preceding sequence, a subsequent dense layer is introduced, featuring 64 neurons that are characterized by the utilization of ReLU activation. The final layer of classification is composed of a dense layer with 10 neurons and graced by the presence of a softmax activation function. Its design is specifically optimized for the intricacies of multi-class classification tasks, and it is enhanced by the utilization of a softmax activation function. The activation function assigns probability scores to each class, and the class with the greatest score is designated as the predicted class for the input image.

The ResNet model incorporates residual connections within each residual block, allowing the input of the block to be preserved and propagated to the output. The use of this technique enables a connection between the initial depiction of the input image and the more profound layers of the model (Xie et al., 2017). The persistence of the effect of color space modification on feature extraction from images has been demonstrated (Oyedotun, Al Ismaeil & Aouada, 2022). Additionally, this technique guarantees that the model’s extraction of deeper features from images in various color spaces exhibits a higher degree of unpredictability, hence mitigating the repetitive nature of the feature maps. This has the potential to provide additional feature patterns to the network and enhance the benefits of utilizing images in various color spaces.

The testing of the model GCNN begins with the implementation of image classification on the original PlantVillage dataset. It is done by feeding the input images in its unprocessed RGB format, yielding results that are undeniably remarkable. The model’s exceptional performance is evident in its classification accuracy of nearly 98 percent. The duration allocated to this undertaking totals a modest 14 s, serving as evidence of the model’s effectiveness. With a constant dedication to exploration, we propel our investigation forward by undertaking an image classification with the utilization of color space conversion techniques. The chosen transformation leads us into the vibrant realms of HSV, LAB, YCrCb, YUV, LUV, XYZ, and HLS color spaces. The selection of these color spaces was not arbitrary but driven by pragmatic considerations, primarily rooted in the simplicity and robustness of performing the conversion using the versatile Scikit-learn library. The results of this multifaceted classification odyssey have been meticulously documented in Table 1, offering insights into the variances introduced by these diverse color spaces.

Table 1 Comparison of color spaces on plantvillage.

Color space	Execution time	Accuracy	Cross entropy loss	F1-score	
RGB	1 h 26 m 16 s	97.88	6	1.0	
HSV	1 h 02 m 15 s	99.32	3	0.93	
LAB	1 h 12 m 15 s	97.67	2	0.88	
YCrCb	1 h 09 m 44 s	99.22	1	0.85	
XYZ	1 h 01 m 23 s	96.29	7	0.88	
HLS	1 h 45 m 18 s	96.90	3	0.92	
LUV	1 h 27 m 57 s	99.40	2	0.93	
YUV	1 h 51 m 11 s	99.42	1	0.86	

Dataset description

A few years ago, a virtual open-access platform was created to learn about crop health and diseases. This platform was named PlantVillage (available at www.plantvillage.org). It has a database of data on more than 1,800 illnesses of more than 150 crops. Experts in plant pathology have penned this content, which incorporates data from academic publications. Every image included in the PlantVillage database was taken at experimental research stations run by US Land Grant Universities, including Penn State, Florida State, Cornell, and others. The expert plucks the leaves and collects them from field trials of crops infected by some disease. The leaves were then positioned on top of a sheet of article with a black or grey background. Every image was shot outdoors in broad daylight. The said dataset contains 54,309 healthy and unhealthy leaf images collected from 14 different crop species. Further, these images are divided into 38 categories by species and disease. For this study of binary classification, we have taken only 9,446 healthy and unhealthy images of 10 classes from PlantVillage. Here, the dataset does not have any issue of class imbalance as it contains 5,163 healthy and 4,283 unhealthy leaves. This subset of the dataset is divided into three sets: the training set with 70% of data, the validation set with 10% of data, and the test set with 20% of data. The proposed model is implemented under Tensorflow. We have set up the training options and hyper-parameters for the proposed model. The model training and testing for the said dataset is done using Anaconda Distribution, which is a general-purpose Python notebook. This is one of the common tools used to perform deep learning, data visualization tasks, and many others. The configuration of the computing machine is as follows: Workstation with Windows-10 operating system, NVIDIA GeForce 8 GB GPU, Intel i7 CPU @ 4.20 GHz, and 32 GB (DDR4) RAM.

Results and discussion

The objective of our research work is to examine the influence of various color spaces on the precision of crop disease detection. We conducted a comprehensive investigation, utilizing a range of color spaces such as HSV, LAB, YCrCb, XYZ, HLS, LUV, YUV, and the widely used RGB. This detailed section comprehensively analyses the accuracy of the results obtained with each color space. These findings provide valuable insights into the potential of these color spaces for crop disease classification. The HSV color space, an intricate model that separates hue, saturation, and value, emerged as a standout performer in crop disease detection. With an accuracy rating of 99.32%, the HSV color space demonstrated its prowess in capturing and distinguishing even the most subtle color variations that serve as key indicators of crop diseases (Fig. 3). The significance of this achievement cannot be overstated, as it underscores the importance of considering the HSV color space in addressing the complex challenges of agricultural image analysis. HSV’s capability to dissect the color composition of images, untangling it from luminance and intensity variations, has made it a compelling choice.

Figure 3 Model loss and accuracy for HSV color space.

LAB, a color space that characterizes colors based on luminance and the two color opponent channels, a* and b*, demonstrated a commendable accuracy of 97.67 percent, as shown in Fig. 4. What makes LAB particularly intriguing is its ability to perceptually mimic human vision. The ‘L*’ channel, representing luminance, stands out for its importance in detecting disease-related variations in image intensity. Meanwhile, the ‘a*’ and ‘b*’ channels delve into the chromatic attributes of an image, capturing the subtleties in color that are vital for disease identification. The ability of LAB to effectively encapsulate both luminance and color changes positions it as a valuable tool in the crop disease detection arsenal. The YCrCb model excels in untangling luminance (Y) from chrominance (Cr and Cb) information, making it a versatile choice for capturing disease-related color variations in crop images. The Y channel encapsulates the image’s brightness and intensity, while the Cr and Cb channels delve into the chromatic properties. The observed high accuracy underscores the potential of YCrCb in enhancing disease detection through a robust separation of these attributes. Our experimentation with the YCrCb color space, commonly employed in video and image compression, yielded an impressive accuracy of 99.22 (Fig. 5). However, the loss still jumps around towards the end of the training, which does not suggest the fact that the model has picked up any useful features. This observation proved that the model is failing to some extent to learn anything.

Figure 4 Model loss and accuracy for LAB color space.

Figure 5 Model loss and accuracy for YCrCb color space.

The XYZ color space, recognized for its linear and device-independent color representation, provided an accuracy of 96.29 percent, which is shown in Fig. 6. Although this slightly trailed some other color spaces in terms of accuracy, it showcased its ability to encode essential disease-related information within its channels. The XYZ model inherently segregates color information from luminance, offering a balanced framework for capturing intricate disease-induced color patterns. Alternatively, it can be observed from Fig. 6 based on the validation loss and accuracy curves that the model is not learning anything as curves are showing too much gap loss, accuracy, and validation. Here, hyperparameter tuning can be required. blackHLS, known for segregating hue, lightness, and saturation attributes, achieved an accuracy rating of 96.90 percent (Fig. 7). These results underscore the capacity of HLS to quantify crop diseases in terms of color and lightness. This distinctive feature allows it to effectively capture variations in color tone and intensity that may be indicative of different diseases. HLS may serve as a compelling choice for scenarios where both color and brightness information play significant roles in disease identification.

Figure 6 Model loss and accuracy for XYZ color space.

Figure 7 Model loss and accuracy for HLS color space.

LUV, a color space designed to mimic human vision, delivered an exceptional accuracy of 99.40 percent (Fig. 8). The LUV model excels in capturing subtle color changes that may serve as strong markers of crop diseases. The ‘L*’ channel, similar to the LAB color space, addresses luminance, while the ‘u*’ and ‘v*’ channels delve into the chromatic nuances. The high accuracy of LUV underlines its efficacy in modeling disease-induced color shifts and positions it as a potent candidate for crop disease identification. Our exploration into the YUV color space, recognized for its compatibility with video transmission, culminated in a noteworthy accuracy of 99.42 percent (Fig. 9). YUV, similar to YCrCb, effectively decouples luminance (Y) from chrominance (U and V), offering a dynamic transformation of color information. This transformation lends itself to capturing subtle color variations related to crop diseases. The high accuracy achieved in YUV highlights its potential to enhance disease detection through the unique separation of these attributes. RGB, the most prevalent color space and the native representation for digital images achieved a respectable accuracy of 97.88 percent (Fig. 10). While it remains a competitive choice for crop disease detection, our results indicate that alternative color spaces, such as HSV and LUV, may offer a marginal edge in terms of accuracy. Nevertheless, RGB still provides a strong foundation for disease identification.

Figure 8 Model loss and accuracy for LUV color space.

Figure 9 Model loss and accuracy for YUV color space.

Figure 10 Model loss and accuracy for RGB color space.

The extensive variability in accuracy results across different color spaces brings to light the intricate role of color representations in the domain of crop disease detection. This comprehensive analysis allows us to draw several noteworthy conclusions and implications. First and foremost, it is imperative to acknowledge that there exists no universal color space that is optimal for all crop disease detection scenarios. Each color space exhibited a unique set of strengths and limitations, underscoring the necessity to carefully consider the specific attributes of the crop diseases under examination. The optimal choice of color space should be contingent upon the nuanced characteristics of the agricultural images and the diseases in question.

Secondly, the remarkable accuracy achieved in YCrCB, HSV, LUV, and YUV color spaces reaffirms their substantial significance in the realm of agricultural image analysis. These color spaces emerged as highly effective in capturing the intricate and subtle color patterns that signify crop diseases. Their performance highlights their potential to elevate the accuracy of disease identification systems and, by extension, enhance crop management practices and mitigate the risk of crop losses. Additionally, it is crucial to emphasize that certain color spaces, including LAB HLS and XYZ, while not securing the highest accuracy ratings, continue to offer valuable insights for crop disease detection. These color spaces provide unique perspectives on the representation of disease-induced color variations, thereby contributing to a more comprehensive understanding of the detection process. In specific scenarios, these color spaces remain viable choices, depending on the nuances of the diseases and the corresponding images.

It summarizes that the selection of an appropriate color space is a critical decision in the context of crop disease detection. By strategically choosing and harnessing specific color spaces based on the nature of the agricultural images and the diseases under scrutiny, it is feasible to amplify the accuracy of disease identification systems. These insights hold significant promise for improving crop management practices, reducing crop losses, and, ultimately, making substantial contributions to global food security and sustainable agriculture.

Conclusion

In this study, we investigate the influence of various color spaces on the classification of diseased images. Here, Eight distinct color spaces, namely RGB, HSV, LAB, YCrCb, XYZ, HSL, LUV, and YUV are examined in the context of crop disease classification tasks. With this investigation, comprehensive guidelines are provided for the successful implementation of classification tasks in the field of crop disease detection and classification. The PlantVillage database is utilized for the purpose of evaluating the model in this particular implementation. For this investigation, we designed the GCNN model, which is capable of being trained once to achieve classification accuracy for images in many color spaces on subsequent tasks. The utilization of group convolution in the design principle of the GCNN model is employed as a means to tackle the issue of training decoupling. Our research work highlights the significance of carefully choosing a suitable color space in the context of crop disease detection. It is possible to enhance the accuracy of disease identification systems by intentionally selecting and utilizing certain color spaces that are appropriate for the agricultural imagery and diseases being examined. The aforementioned observations with RGB, HSV, XYZ, HSL, and LUV exhibit considerable potential in enhancing agricultural methods, mitigating crop losses, and ultimately playing a big role in advancing global food security and sustainable agriculture. Our experimentation with the LAB, YCrCb, and YUV color space yielded impressive accuracy. However, the loss still jumps around towards the end of the training, which does not suggest that the model has picked up any useful features. This observation proved that the model is failing to some extent to learn anything.

To effectively integrate feature information from various color spaces, we are planning to employ the channel shuffle operation to facilitate fusion between different convolutional groups. It would help us to improve the accuracy of the model for images containing various colors. The investigation of the impact of a single channel from any color space in the channel shuffle operation for the subsequent classification work allows us to examine whether the inclusion of the color space diminishes the ranking of the remaining channels.

Supplemental Information

Supplemental Information 1 Source Code of Color space-based crop disease classification using GCNN.

Additional Information and Declarations

Competing Interests

Author Contributions

Data Availability

Khalid Raza is an Academic Editor for PeerJ Computer Science.

Naeem Ahmad conceived and designed the experiments, performed the experiments, analyzed the data, prepared figures and/or tables, authored or reviewed drafts of the article, and approved the final draft.

Shubham Singh conceived and designed the experiments, performed the experiments, analyzed the data, performed the computation work, authored or reviewed drafts of the article, and approved the final draft.

Mohamed Fahad AlAjmi performed the experiments, analyzed the data, performed the computation work, authored or reviewed drafts of the article, and approved the final draft.

Afzal Hussain performed the experiments, analyzed the data, performed the computation work, authored or reviewed drafts of the article, and approved the final draft.

Khalid Raza conceived and designed the experiments, analyzed the data, prepared figures and/or tables, authored or reviewed drafts of the article, and approved the final draft.

The following information was supplied regarding data availability:

The PlantVillage Dataset is available at Kaggle: https://www.kaggle.com/datasets/emmarex/plantdisease.

The code is available in the Supplemental File.

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
