# Peer review of "CropGCNN: color space-based crop disease classification using group convolutional neural network"

_PeerJ Computer Science, doi:10.7717/peerj-cs.2136_

## Round 0.1 · original submission · Minor Revisions

The paper can be accepted with minor revision.

·

Basic reporting

The paper uses clear and professional English throughout. Literature references and sufficient field context are provided. The professional article structure, figures, and tables are well-presented. The research is self-contained and the results are relevant to the hypothesis.

Experimental design

The paper presents original research within the journal's aims and scope. The research question is well-defined, relevant, and meaningful. The methods section provides good detail, but could be improved by including justification for choosing 3 as the optimal "groups" parameter value (lines 140 and 161-162).

Validity of the findings

Since the dataset is publicly available, replication should be straightforward. The conclusions are well-stated but could be strengthened by including comparisons to baseline models for context.

Additional comments

The paper demonstrates a promising GCNN model with multiple color spaces, but it would benefit from addressing the following key points:

1. Justification for choosing 3 as the optimal "groups" parameter: The authors should explain why they chose this value and how it compares to other choices in the "Results and Discussion" section.
2. Inclusion of baseline models: The discussion should include comparisons to existing state-of-the-art image classification models to highlight the proposed model's relative performance.
3. Assessment of impact and novelty: The authors should discuss the impact and novelty of their findings, emphasizing how this research contributes to the field.

By addressing these points, the authors can strengthen the paper and increase its potential for publication.

Reviewer 2 ·

Basic reporting

I think the underlying idea of this work is very interesting, seeking to use multiple color spaces than just single RGB color space as additional inputs. The authors also have done a great job introducing prior literature and articles. In terms of the figures and tables, I think there're a few typos and misses authors could make improvements on. For example, in table 1, the first row, I believe the authors meant 16 seconds, and in the loss vs epochs plot, what is the loss, is it a cross entropy loss or log loss? Not clear, I think providing such information is important for reviewers to understand the context. The position of plot titles are not in a consistent location either, making a bit difficult to read. Also, what does the underlying dataset contain? For example, the sample size, number of labels to be classified? Does the time in table I mean the entire training time? These background information is important even though they're from an already known dataset. This just gives readers more clarity on the problem this research work is focusing on.

Experimental design

In terms of the experimental design and let's set aside of validity of the results for a moment, the authors should show at least experimentations with multiple datasets to support their hypothesis. The authors think by purposely selecting and combing certain color spaces can improve model performance, which I think it is a very interesting proposal. However, this should be shown on a variety datasets, and not just one dataset. If the same trend can be found, I would feel much more confident about this idea.

Validity of the findings

I specifically have questions about the validity of figure 5 and 6. Based on the validation loss curves, I don't think the model is learning anything. The loss still jumps around towards the end of the training, which does not suggest the fact that model has picked up any useful features. I think the authors have to go through some hyperparameters tuning to improve it.

---

## Round 0.2 · Minor Revisions

The authors must improve the paper just looking to the observations of the reviewer,

Reviewer 2 ·

Basic reporting

All previous concerns have been addressed.

Experimental design

The authors have provided more information on the underlying dataset, which is good. However, som questions still remains. For example, what kind of classification task is it? Is it a binary classification on healthy vs non-healthy plants? Is there any imbalance issue in the dataset?

Validity of the findings

Authors have corrected some of the concerns on figure 5 and 6 based on previous iteration. I think the conclusion should be changed accordingly as well. Clearly, the color spaces, LAB, YCrCb and YUV did not provide any useful information given their poor performance. This shows only a selective of color spaces has predictive power and this should be reflected in the conclusion (e.g., line 300)

---

## Round 0.3 · accepted · Accept

The paper was well improved. The authors responded to all minor comments.